



# 1 An internal solitary wave forecasting model in the northern South
# 2 China Sea (ISWFM-NSCS)

Yankun Gong[1], Xueen Chen[2], Jiexin Xu[1], Jieshuo Xie[1], Zhiwu Chen[1], Yinghui He[1], Shuqun Cai[1,3,4]
[1]State Key Laboratory of Tropical Oceanography, South China Sea Institute of Oceanology, Chinese Academy of Sciences,
Guangzhou 510301, China
[2]College of Oceanic and Atmospheric Sciences, Ocean University of China, Qingdao, 266100, China
[3]Institution of South China Sea Ecology and Environmental Engineering, Chinese Academy of Sciences, Guangzhou 510301,
China
[4]University of Chinese Academy of Sciences, Beijing, 100049, China
*Correspondence to*: Shuqun Cai (caisq@scsio.ac.cn)
**Abstract.** Internal solitary waves (ISWs) are a ubiquitous phenomenon in the dynamic ocean system, which play a crucial role
in driving transport through turbulent mixing. Over the past few decades, numerical modelling became a vital approach to
investigate the generation mechanism and spatial distribution of ISWs. The northern South China Sea (NSCS) has been treated
as a physical oceanographic focus of ISWs in massive numerical studies since last century. However, there was no systematic
evaluation of a reliable three-dimensional model about accurately reproducing ISW characteristics in the NSCS. In this study,
we implement a three-dimensional ISW forecasting model in the NSCS and quantitatively evaluate the requirements of factors
(i.e., model resolution, tidal forcing, and stratification selection) in precisely depicting ISW properties by comparing with
observational data at a mooring station in the vicinity of the Dongsha Atoll. Firstly, the 500 m-resolution model can basically
reproduce the principal ISW characteristics, while the 250 m-resolution model would be a better solution to identify wave
properties, specifically increasing 40% accuracy of predicting characteristic half-widths. Nonetheless, a 250 m-resolution
model spends nearly fivefold computational resources of a 500 m-resolution model in the same model domain. Compared with
the former two, the model with a lower resolution of 1000 m severely underestimates the nonlinearity of ISWs, resulting in an
incorrect ISW field in the NSCS. Secondly, the model with eight (or thirteen) primary tidal constituents can accurately
reproduce the real ISW field in the NSCS, while the one with four main harmonics (M2, S2, K1 and O1) would underestimate
averaged wave-induced velocity for about 38% and averaged mode-1 wave amplitude for about 15%. Thirdly, the model with
the initial condition of field-extracted stratification gives a better performance in predicting some wave properties than the
model with climatological stratification, namely 13% improvement of arrival time and 46% improvement of characteristic
half-width. Finally, background currents, spatially varying stratification and external (wind) forcing are discussed to reproduce
a more realistic ISW field in the future numerical simulations.





## 1 Introduction

Numerical simulations, one of the most important approaches to investigate internal solitary waves (ISWs) in the world's oceans, have been gradually developed from two-dimensional (e.g., Du et al., 2008; Buijsman et al., 2010) to three-dimensional (e.g., Zhang et al., 2011; Alford et al., 2015) over the past few decades. South China Sea (SCS), the largest marginal sea in the northwest Pacific, has been commonly known as an active region of ISWs via massive in-situ observations (cf. Ramp et al., 2004, 2019; Farmer et al., 2009, 2011) and numbers of remote sensing images (cf. Liu et al., 2004; Zheng et al., 2001, 2007). Although the vertical structure and horizontal distribution on the sea surface of ISWs can be nicely illustrated by field measurements at sparse sites and satellite images, respectively, they are still of limited value for telling a complete story of ISWs in the entire northern SCS (NSCS). Complementary to in-situ and remote-sensing observations, numerical model can give a comprehensive characterization in the ISW field in case of realistic initial and boundary conditions. Hence, we take NSCS as an example to introduce a high-performance ISW forecasting model and quantitatively evaluate requirements of model configurations (i.e., resolution, tidal forcing, and stratification selection) for accurately reproducing a real ISW field.

With the development of higher performance computing facilities, a variety of three-dimensional (3D) realistic numerical models with structured and unstructured grids were established for simulating ISWs in the NSCS (see Table 1), such as MITgcm (Vlasenko et al., 2010), SUNTANS (Zhang et al., 2011) and FVCOM (Lai et al., 2019). Meanwhile, the model capabilities have been continuously improved (Simmons et al., 2011). Specifically, the model resolution was effectively enhanced from 250-1000 ($\Delta x$-$\Delta y$) m (Guo et al., 2011) in a limited domain to 150/300 m in a large domain including the entire NSCS (Zeng et al., 2019). From past to present, the barotropic tidal forcing dataset TOPEX/Poseidon Solution (TPXO, Egbert and Erofeeva, 2002) and climatological stratification dataset World Ocean Atlas (WOA, Locarnini et al., 2018) have been updated with higher resolutions both in horizontal and vertical, providing more realistic and precise boundary and initial conditions in the model configurations. Although it is commonly known that a higher-resolution model can tell a more complete story of ISWs, the usage of computational resources is worthy to be considered. Thus, what resolution of model is needed to give an accurate depiction of ISW fields and simultaneously save the computational cost is still a question.

Even though numbers of previous in-situ observations have shown the four barotropic tidal constituents (M2, K1, O1 and S2) are dominant at the Luzon Strait (Zhao and Alford, 2006; Farmer et al., 2009), the other barotropic tidal constituents (e.g., N2, K2, P1 and Q1) are also non-negligible (Beardsley et al., 2004). Historically, numerical simulations with different numbers of tidal constituents have been widely employed to investigate the physical dynamics of ISWs in the NSCS, i.e., single K1 harmonic (Li, 2014), four tidal harmonics (Buijsman et al., 2010), and eight primary tidal harmonics (Alford et al., 2015; Jin et al., 2021). Among these, eight tidal constituents were most commonly applied in the 3D models. However, other tidal constituents, such as M4, MS4, MN4, MM, and MF, were yet to be considered. The questions that arise are whether a single tidal constituent can satisfy the reproduction of a real ISW field and how many tidal constituents are required for running an accurate 3D realistic ISW model.





Apart from resolution and tidal forcing, stratification selection is also an important factor in improving model accuracy. A
horizontally homogenous stratification profile was normally implemented as an initial condition in a 3D realistic model (cf.
Zhang et al., 2011; Lai et al., 2019). Specifically, a domain average of the climatological dataset (WOA) is one of the most
common options (Vlasenko et al., 2010; Zeng et al., 2019), since the in-situ observational data are relatively inaccessible. Once
the field data at an isolated mooring station are available, are they a better choice than the climatological data to be the model's
initial condition? What if the mooring is near-field (in the vicinity of the Luzon Strait, the ISW generation site) or far-field
(e.g., in the deep basin or over the continental slope and shelf)?
In this paper, we attempt to introduce a high-performance ISW forecasting model and evaluate the roles of different
resolutions, initial and boundary conditions in accurately reproducing ISWs via a series of sensitivity 3D non-hydrostatic
numerical simulations. The paper is structured as follows. In section 2, configurations of the 3D forecasting model are
introduced, as well as the simultaneous remote sensing images and in-situ observations. The model calibrations are presented
in section 3. In section 4, we quantitatively illuminate the requirements of model resolutions, tidal constituents, and initial
stratification selection for a reliable 3D ISW forecasting model. Discussion and conclusions follow in section 5.

## 75  2 Data and Methods

To characterize the real ISW field in the NSCS, we implement an ISW forecasting model (ISWFM-NSCS) and compare the
modelled wave properties on the continental slope with those observed at in-situ mooring station DS (marked as magenta star
in Fig. 1a). Remote sensing images are downloaded for the model calibration as well.

### 79  2.1 Numerical modelling

Although running a 2D slice model is much more economical than running a 3D model from the perspective of
computational resources, the 2D model cannot correctly reproduce the ISW field in the real ocean (see Appendix A). Therefore,
we implement a realistic 3D non-hydrostatic primitive equation ocean solver (MIT general circulation model, MITgcm,
Marshall et al., 1997) in the spherical coordinate to reproduce the ISWs features in the NSCS. The model domain (115.8° –
123.8°E, 17.8° – 22.3°N, see blue box in Fig. 1a) includes the main generation site of ISWs (i.e., Luzon Strait) and the mooring
station DS on the continental slope. Bathymetry data are derived from the global gridded bathymetry dataset GEBCO
(https://www.gebco.net/data_and_products/gridded_bathymetry_data). To keep the consistency with the instrumental
deploying period, we start the model from 00:00 UTC 5 August 2014 and last fifteen days. Previous statistical analyses, based
on SAR images in the NSCS from 1995 to 2001, also indicated that ISW occurrence frequencies were relatively high in August
(Zheng et al., 2007). The initial model temperature and salinity profiles (see black and blue lines in Fig. 1b) are derived from
the WOA18 climatology dataset (World Ocean Atlas 2018) by spatially averaging the monthly output in August, resulting in
horizontally-uniform conditions. Density and buoyancy frequency profiles are shown as black lines in Figs. 1c and 1d.





To ensure ISWs can be physically derived and consider the computational efficiency, the horizontal cell ($\Delta x$) is set as 500
m in both zonal and meridional directions. In order to satisfy the high-mode vertical resolution requirements, 90 vertical layers
are spaced in accordance with the hyperbolic tangent function (Stewart et al., 2017), namely ranging from 5 m near the surface
to 120 m near the sea bed (in the deep water). We impose a time step of $\Delta t = 10$ sec to satisfy the Courant-Friedrichs-Lewy
(CFL) conditions in both horizontal and vertical. The Coriolis parameter is constant in the entire model domain, which is given
by a value at a latitude of $20.5°$ ($f = 5.1 \times 10^{-5}$ rad s$^{-1}$). To determine whether non-hydrostatic mode is necessary, we also run
a hydrostatic model (not shown). It is noting that fake internal solitary-like wave trains, also called spurious non-hydrostatic
processes (Alvarez et al., 2019), are clearly visible at first glance, suggesting that hydrostatic mode is inappropriate for a high-
resolution model of ISWs. We therefore configure the model in non-hydrostatic mode.
The control run (Exp. 1, ***500m_8HARs***) is driven by eight main tidal constituents (M2, S2, N2, K2, K1, O1, P1, and Q1) on
the four open boundaries with values originated from the Oregon State University TOPEX/Poseidon Solution (TPXO8-atlas
data) with 1/30º resolution (Egbert and Erofeeva, 2002). A 25 km wide sponge layer is imposed on each lateral boundary to
absorb internal wave energy and avoid wave reflection back to the inner region. Quasi-steady conditions occur after 3 days, so
the model results are analyzed over the remaining 12 days (8 – 20 August). The control run (***500m_8HARs***) runs at 1 h time
interval in the entire model domain and single-point outputs with a higher sampling rate of 1 min at the selected station DS for
recording the local ISW properties, and thereby comparing to the in-situ observations. Constant horizontal and vertical eddy
viscosity and diffusivity coefficients are imposed as $A_h = 0.5$ m$^2$ s$^{-1}$; $A_v = 5 \times 10^{-3}$ m$^2$ s$^{-1}$; $K_h = 0.5$ m$^2$ s$^{-1}$; $K_v = 5 \times 10^{-3}$
m$^2$ s$^{-1}$ to eliminate grid-scale instability (Legg and Huijts, 2006). The bottom stresses are parameterized using a quadratic law
with a bottom drag coefficient of $C_d = 2.5 \times 10^{-3}$.
**2.2 Remote sensing images**
Remote sensing imagery contains lots of detailed information of ISW properties, including wave crest lines and their arrival
time, which was commonly applied in the NSCS (Liu et al., 2004; Zheng et al., 2007). Here we download two MODIS true-
color pictures with a horizonal resolution of 250 m at 05:15 UTC on 14 August and at 02:50 UTC on 15 August 2014,
respectively. In addition, we compute the horizontal gradients of sea surface height ($|\nabla\eta|$, in the unit of cm km$^{-1}$), which detects
the variations in surface roughness caused by the ISW-induced convergent and divergent currents, thereby producing
analogous images to the satellite images. Note that the model is hourly sampled, so we select the closest snapshots of $|\nabla\eta|$ at
05:00 UTC on 14 August and at 03:00 UTC on 15 August 2014 to compare with MODIS images.
**2.3 In-situ measurements**
The through-water-column mooring station DS (magenta star in Fig. 1a) is located at 117º44.7'E, 20º44.2'N in the vicinity
of the Dongsha Atoll, which was deployed at a water depth of ~1250 m from 1 August to 28 September 2014. Three acoustic
Doppler current profilers (ADCPs) measured currents ranging from a depth of 1180 m to the sea surface every two minutes





with 16-m vertical bins in upper 900 m and 8-m vertical bins below 900 m. The mooring was configured by temperature
sensors, conductivity-temperature-depth (CTD) sensors and conductivity-temperature (CT) sensors at different water depths.
The temperature sensors were at 10 m, 30 m, 50 m, 90 m, 130 m, 150 m, 170 m, 250 m, 350 m, 500 m, 600 m, 700 m, 800 m,
950 m, 1050 m, and 1220 m; the CTD sensors were at 1 m and 1100 m; and the CT sensors were at 20 m, 40 m, 70 m, 110 m,
150 m, 200 m, 300 m, 450 m, 550 m, 650 m, 750 m, 850 m, 1000 m, and 1200 m. Temporal sampling rates were 10 sec for
the temperature and CTD sensors, and 15 sec for the CT sensors, respectively. The instruments carried by the moorings
generally functioned well, but CT sensors stopped working after 6 September 2014 due to the lack of power. Besides, Xu et
al. (2020) indicated that an anti-cyclonic eddy dominated the region of the mooring since mid-September 2014, which
significantly affected the local wave properties at the DS station. To avoid the impacts of background currents, we selected
fifteen ISWs during the spring tidal period from 00:00 UTC 8 August to 00:00 UTC 15 August as criteria to quantitatively
evaluate the performance of sensitivity numerical experiments.
**3 Model results and calibrations**
In this section, we validate the model accuracy from three aspects: barotropic tidal constituents via comparing with TPXO8-
atlas dataset and in-situ observational data; spatial distributions of ISWs via comparing with the remote-sensing images; wave
properties (i.e., amplitude, arrival time, wave-induced velocity and propagation direction) of ISWs via comparing with the in-
situ observations at mooring station DS.
**3.1 Barotropic tide calibrations**
The 3D control run only runs for 15 days, which is too short to do the harmonic analysis. To validate the model accuracy in
simulating the barotropic currents of eight key tidal constituents, we rerun a 3D model (Exp. 2, ***500m_8HARs_BT***) with the
same configurations as ***500m_8HARs***, but extend the duration time to 100 days and turn off the iteration of temperature and
salinity to focus on the barotropic tide regimes.
As M2, S2, K1, and O1 barotropic tides are dominant in the NSCS (Ramp et al., 2004; Farmer et al., 2009), here we calculate
the amplitude ($U$) and phase ($\phi$) of the zonal velocity ($\boldsymbol{u_{bt}}$) by doing the harmonic analysis over the last 90 days in Exp. 2
(***500m_8HARs_BT***) and compare them with the TPXO8-atlas dataset. A root-mean-square error ($RMSE$), referring to
Cummins and Oey (1997), is computed to evaluate the model performance in the barotropic regime, which is given by
$$RMSE_h = \sqrt{\frac{1}{2}[(U_m^2 + U_o^2) - U_m U_o \cos(\phi_m - \phi_o)]}, \qquad (1)$$

in which, subscript $h$ represent four different harmonics; $U$ and $\phi$ are amplitude and phase of zonal barotropic currents with
the subscripts $m$ for model and $o$ for observation (TPXO8-atlas). We therefore obtain the horizontal distributions of $RMSE$
for four tidal constituents (see Figs. 2a – 2d). In most model domain, $RMSE$ is less than 0.02 m s$^{-1}$, but slightly larger in the



shallow water (e.g., Luzon Strait and the continental shelf), which is still less than 0.2 m s$^{-1}$. It may be because that the
bathymetry derived from the GEBCO dataset and resolutions in our model differ from those in the TPXO8-atlas, thereby
resulting in the discrepancy.
In addition to comparison between this model and the global tide model, we extract the DS station outputs with a high
sampling rate for comparing with the in-situ observations. To avoid the effects of massive high-frequency motions (i.e.,
environmental noises) in the observational time series on the barotropic regime, we first do the harmonic analysis for zonal
barotropic velocities from 5 August to 19 September, then extract the amplitude and phase of eight key tidal constituents, and
restructure the time series (see red line in Fig. 2e). In terms of the model results, we obtain the time series at station DS in the
same way (see black line in Fig. 2e). It is worth mentioning that the discrepancy between the eight-harmonic restructured time
series and the raw data in the model is small, since the experiment is basically driven by the eight tidal constituents and does
not include any affects from the background environment. By comparing the two timeseries, the model reliability is validated
all through the spring and neap tides. Overall, the model presents nice performance in the barotropic regime.

**3.2 Comparison with MODIS images**

Apart from the model validation in barotropic tides, we then look over the control run (***500m_8HARs***) in baroclinic (ISWs)
regime by comparing the model results with MODIS images. Figs. 3a and 3b both show two successive ISWs (labeled as IW1
and IW2) in the deep basin with a distance of ~120 km. The lengths, curvatures and locations of IW1 and IW2 in the simulation
are consistent with those in the MODIS image. However, two other ISWs occurring over the continental slope and shelf are
captured in the numerical simulations, but not observed on 14 August in the MODIS-Aqua image due to the cloud covering
(Fig. 3b). Conversely, the cloud disappeared on 15 August, so the MODIS-Terra sensor gives a clear seascape painting of
ISWs both in the shallow water (i.e., IW2) and deep water (i.e., IW3 and IW4). Note that IW2 in Figs. 3b and 3d are the same
ISW, which propagates ~250 km within 19 hours and 35 mins. All ISWs (IW2, IW3 and IW4) in Figs. 3c and 3d occur at the
fairly close locations with analogous wave properties. From the perspective of crestline lengths, the numerical model shows
well agreement with the MODIS images, namely 131 km versus 133 km for IW2 in Figs. 3a and 3b; 187 km versus 198 km
for IW3 and 74 km versus 69 km for IW4 in Figs. 3c and 3d. Besides, in the water depth shallower than 500 m, the modelled
IW2 exhibits an ISW train with trailing waves, which is also shown in the MODIS image. As the model neglects wind above
the sea surface and other marine dynamical processes, there are still some subtle nuances of wave characteristics between them.
Overall, this model nicely demonstrates spatial distributions of ISWs in the NSCS, based on the comparison with remote
sensing imagery.

**3.3 Comparison with in-situ observations**

To further evaluate the model performance in reproducing ISWs, we introduce the in-situ observations. The vertical structure
and timing of the wave arrivals, after crossing the deep basin, can be seen in details using daily plots (Figs. 4a – 4g) of the
temperature isotherms and baroclinic (ISW-induced) velocities from 8 to 14 August at mooring DS. For clarity, only the results



in upper 900 m are shown in Fig. 4, including the main wave-induced temperature fluctuations. Alford et al. (2010) suggested
that the nonlinear internal wave speeds varied from 2.0 m s$^{-1}$ near the Luzon Strait to 3.0 m s$^{-1}$ in the deep basin in the NSCS,
so it takes roughly two days for ISWs to propagate from the generation site to the targeted station (DS). We move the arrival
time (i.e., 8 to 14 August) of ISWs two days forward at the station DS, so the related barotropic tides gradually increase during
the spring tidal cycle at the Luzon Strait (i.e., 6 to 12 August). It explains why ISWs were relatively weak and linear from 8 to
10 August (Figs. 4a – 4c), but became significant and nonlinear from 11 to 14 August (Figs. 4d – 4g). A single ISW was
captured around 12:00 UTC from 11 to 14 August, which arrived at the location at approximately the same time every day
(termed as type-a ISWs by Ramp et al. (2004)). Meanwhile, a wave train, consisted of two dominant solitons and some small
trailing waves, arrived at the station an hour later each day, showing the same wave characteristics as type-b ISWs in Ramp et
al. (2004).
In terms of the model, we also use the daily plots (Figs. 4h – 4n) at station DS with 1 min sampling rate to show its similarity
to the in-situ observations. An increasing trend of wave amplitude and nonlinearity is obvious from 8 August to 14 August in
the model results, suggesting precise depictions of barotropic tides and ISWs' characteristics. Specifically, both type-a (single
solitons) and type-b ISWs (wave trains) are displayed with analogous arrival time, wave-induced (baroclinic) velocity (color
shades in Fig. 4) and wave amplitude (contours in Fig. 4) to those in the observations. It's worth noting that even the linear
internal tides and/ or hydraulic jumps around 12:00 UTC from 8 to 10 August are reproduced. Although the model omits some
small wave signals (see blue arrows in Fig. 4e) in the observations, which might be induced by non-tidal processes such as
background currents, the model still shows a well performance in the ISW reproduction.
To quantitatively identify the model accuracy, we select fifteen ISWs (marked as red arrows in the left column of Fig. 4),
extract their wave properties (i.e., arrival time, maximum wave-induced velocity, propagation direction and maximum mode-
1 wave amplitude) and compare them between in-situ observations and numerical simulations. In terms of wave propagation
direction, we obtain by computing the angle of baroclinic zonal and meridional components in the layer with maximum velocity.
The maximum mode-1 wave amplitude ($A_1$) is extracted from the mooring data and model outputs by least squares fitting
density perturbation profiles $\rho'(z)$ to normalized modal structure function $W_n(z)$, following the similar procedures to those
described by Buijsman et al. (2010) and Rayson et al. (2012). Although the mode-1 wave amplitude can also be extracted by
least squares fitting the horizontal baroclinic velocity, Rayson et al. (2019) suggested that the method in velocity field was
fuzzy with unidirectional internal waves. The modal structure function can be resolved by a shear-free Taylor-Goldstein
equation with the background stratification $N^2(z)$, which is given by
$$\frac{d^2 W_n(z)}{dz^2} + \frac{N^2(z)}{c_n} W_n(z) = 0, \tag{2}$$

with the boundary conditions $W_n(0) = W_n(-H) = 0$. Subscript $n$ represents the mode number and $c_n$ is the phase speed of
the linear internal waves in $n^{th}$ mode. The buoyancy perturbation $b(z)$, depending on density perturbation $\rho'(z)$, is written as
$$b(z) = -g \frac{\rho'(z)}{\rho_0}, \tag{3}$$





in which, $\rho_0$ is the reference density. Following the internal wave polarization relationships (Gerkema and Zimmerman,
2008), we fit the wave amplitudes ($A_n$) in different vertical modes to $b(z)$ in both in-situ observations and numerical
simulations via

$$b(z) = \sum_{n=1}^{5} A_n N^2(z) W_n(z), \qquad (4)$$

Here, we select the first five vertical modes ($n = 1 - 5$) to do the least squares fitting and mainly discuss the mode-1 wave
amplitude ($A_1$) due to its significant dominance (Fig. 4).
According to the above approaches, we extract the four wave properties for fifteen ISWs and plot Fig. 5, in which
observation and model results are shown in red and green, respectively. First, we list the arrival time of ISWs on the two sides
of Fig. 5. The bias between observation and model is always smaller than 1.5 h and the root mean square deviation (RMSD)
is 0.71 h, indicating accurate depiction of ISW arrival time in the control run (***500m_8HARs***). Second, the maximum baroclinic
velocity (Fig. 5a) and the averaged values (0.98 m s$^{-1}$ and 1.18 m s$^{-1}$, respectively) are shown in the solid lines. It is suggested
that the model underestimates the baroclinic velocity due to neglect of some background non-tidal signals, thereby introducing
a RMSD of 0.41 m s$^{-1}$. Third, the averaged propagation directions of ISWs are ~285º and ~291º, respectively
(the angle measured counterclockwise from north) in the model results and observational data with a RMSD of 8.35º. It is
worth mentioning that the type-a ISWs mainly propagate westward while the type-b ISWs propagate north-westward in both
observation and model, verifying the model's reliability to some extent. Finally, the averaged maximum mode-1 wave
amplitude (~108 m) in the model is close to that (~99 m) in the observation. Nonetheless, the RMSD of mode-1 wave amplitude
is 37.27 m. Overall, the control run can basically reproduce various wave properties of ISWs observed in the vicinity of the
Dongsha Atoll in the NSCS.
**4 Assessment of factors affecting three-dimensional model forecasting precision**
In this section, based on the control run, we alter the model configurations, such as the requirements of horizontal resolutions,
numbers of tidal constituents and initial stratification, to respectively estimate their effects on the model forecasting precision
of ISWs in the NSCS.
To determine the roles of model horizontal resolutions, tidal constituents and initial stratification in reproducing ISWs in
the NSCS, a set of 3D sensitivity numerical simulations are employed with different configurations, which are listed in Table
2. Details in configuration changes are as follows.
1) Exps. 3 and 4 (***250m_8HARs*** and ***1000m_8HARs***): Comparing to ***500m_8HARs***, the horizontal resolution ($\Delta x$) is set as
250 m and 1000 m in both zonal and meridional directions, respectively.
2) Exps. 5 - 7 (***500m_1HAR***, ***500m_4HARs***, and ***500m_13HARs***): Comparing to ***500m_8HARs***, the sensitivity experiments
are driven by single tidal constituent (M2), four main tidal constituents (M2, S2, K1, and O1), and thirteen tidal constituents
(M2, S2, N2, K2, K1, O1, P1, Q1, M4, MS4, MN4, MM, and MF), respectively.





3) Exp. 8 (***500m_Real_N2***): A real stratification profile of background temperature at the mooring station DS is imposed as
the initial condition, which is derived from the in-situ measurements. A backward-in-time low-pass filter derived from a finite
impulse response differential equation is used to compute the background temperature (Rayson et al., 2019).

$$\frac{d\bar{T}}{dt} = \frac{1}{\tau_f}(T - \bar{T}), \tag{5}$$

in which, $\tau_f$ is the filtering time scale, set to 35 h, corresponding to the local Coriolis frequency. $T$ and $\bar{T}$ are the
instantaneous and background temperature, respectively. Then, the background temperature at each observational time step $i$
is given as

$$\bar{T}^{i+1} = \bar{T}^i + \frac{\Delta t}{\tau_f}(T^{i+1} - \bar{T}^i), \tag{6}$$

where $\Delta t$ is the sampling rate (10 secs for the temperature and CTD sensors, 15 secs for the CT sensors). The background
temperature profile is ultimately obtained by low-pass filtering at each layer (see red line in Fig. 1b).

**4.1 Requirements of resolutions**

Various 3D model with different resolutions were implemented to simulate ISWs in the NSCS in previous studies (e.g.,
Vlasenko et al., 2010; Zhang et al., 2011; Lai et al., 2019). However, which resolution is adequate to satisfy the ISW prediction
precision and save computational resources to the utmost in the meantime has yet been discussed. Here, we run two sensitivity
experiments (Exps. ***250m_8HARs*** and ***1000m_8HARs***) with horizontal resolutions of 250 m and 1000 m, to respectively
compare the model performance in different aspects with the control run (resolution of 500 m).
First, the spatial distributions of ISWs are exhibited via the snapshots of sea surface height gradients ($|\nabla\eta|$) at 12:00 UTC
on 12 August 2014. In the control run (***500m_8HARs***), three ISWs (labelled as IWB1, IWA1, and IWB2 from west to east)
with distinct crest lines successively occur between 116ºE and 120ºE (see Fig. 6a), in which IWB1 and IWB2 are internal
wave packets with trailing waves (type-b wave) and IWA1 is a single soliton (type-a wave). As IWB1 approaches the
continental slope and shelf, the leading wave front fully steepens with a narrow characteristic half-width, suggesting its strong
nonlinearity. IWB2 also shows up as a wave packet with many secondary waves in the developing stage, although its
nonlinearity is slightly weaker than IWB1's. Conversely, the single soliton IWA1 with relatively long crest line and broad
characteristic half-width is about to pass mooring station DS (marked as green star in Fig. 6). In comparison, the Exps.
***250m_8HARs*** and ***1000m_8HARs*** reproduce these three waves as well, but with some subtle discrepancies between them. In
Exp. ***250m_8HARs***, more details of wave properties are clarified (Fig. 6b). Specifically, the secondary waves of IWB1 and
IWB2 are more visible than those in ***500m_8HARs***. However, in Exp. ***1000m_8HARs***, some fine structures of ISWs are not
well resolved. For instance, only one secondary wave is found behind the leading wave of IWB2, and the south portion of
IWA1 crest line is barely observed (Fig. 6c).
Then, we select a transect along the main propagation path of ISWs (shown in dashed line in Fig. 6a) at 12:00 UTC on 12
August 2014 to compare the vertical structure of ISWs among three experiments (see Fig. 7). In Fig. 7, blue (yellow) color





shades represent westward (eastward) baroclinic velocity and contours are temperature isotherms. Linear internal waves, such
as internal wave beams near the generation site (120° – 121°E), are nicely reproduced in all numerical experiments. Nonetheless,
nonlinear internal waves present different wave characteristics in different cases. In Exp. **500m_8HARs**, the single soliton
IWA1 and the wave packet IWB2 with a series of trailing waves are apparent in the slice, but IWB1 is not included (Fig. 7a).
In Exp. **250m_8HARs**, IWA1 and IWB2 occur at the same location as those in **Exp. 500m_8HARs**. IWA1 show similar
properties in two cases, but the secondary waves of IWB2 are better described in Exp. **250m_8HARs**. By comparison, IWA1
shows its weak nonlinearity with small vertical displacement and broad characteristic half-width (i.e., horizontal distance
between the wave front and wave trough) in Exp. **1000m_8HARs**. Besides, only one secondary wave appears in the IWB2
packet in Exp. **1000m_8HARs**.

Last, a two-day time segment of observational temperature and baroclinic velocities from 18:00 UTC 11 August to 18:00

UTC 13 August 2014 at the station DS is extracted to demonstrate the sensitivity model capability of simulating vertical
structures of ISWs over the continental slope (Fig. 8). In the control run (**500m_8HARs**, Fig. 8b), two wave packets and two
single solitons successively arrive at the station, keeping the consistency with the observation, although their characteristic
half-widths are slightly broader than those in the field measurements (Fig. 8a). Meanwhile, some small fluctuations, occurring
in the observations, are not included in the control run. In Exp. **250m_8HARs** (Fig. 8c), the half-widths are narrower than
those in the Exp. **500m_8HARs**, which agree better with the real internal wave field. Besides, more fluctuations, i.e., those
small wave signals (09:00 UTC 12 August and 09:00 UTC 13 August) in front of the single solitons are reproduced in this
experiment. Conversely, in Exp. **1000m_8HARs**, internal wave trains can still be reproduced with relatively weak nonlinearity,
but the single solitons are not correct due to their tiny amplitudes and linear wave structures.

To quantitatively evaluate the model performance of sensitivity experiments, we present the bias of five wave properties of

fifteen ISWs (marked as red arrows in Fig. 4) between model results and observational data in Fig. 9. The biases of arrival
time are generally smaller than 1 h (see black and blue circles in Fig. 9a) for Exps. **500m_8HARs** and **250m_8HARs**, whose
RMSDs are 0.71 and 0.67 h, respectively. In contrast, the bias for Exp. **1000m_8HARs** is larger than 1 h (red circles in Fig.
9a) and its RMSD is 0.79 h. In terms of the wave-induced velocity (Fig. 9b), the RMSDs are 0.38, 0.41 and 0.48 m s$^{-1}$ in Exps.
**250m_8HARs**, **500m_8HARs**, and **1000m_8HARs**, respectively. The RMSDs of propagation directions are very close (~8.5°)
in the three experiments (see Table 3). As for the mode-1 wave amplitudes, Exps. **250m_8HARs** and **500m_8HARs**
overestimate the wave amplitudes in most cases (see positive biases in Fig. 9d), thereby resulting in RMSDs of 38.12 and
37.27 m, respectively. Conversely, Exp. **1000m_8HARs** would underestimate the wave amplitudes of majority ISWs with
dominant negative biases in Fig. 9d, resulting in a RMSD of 40.28 m (Table 3). Last but not least, Exps. **500m_8HARs** and
**1000m_8HARs** inaccurately depict characteristic half-widths of ISWs with RMSDs of 1.07 and 2.41 km, while Exp.
**250m_8HARs** performs well with a RMSD of 0.64 km (Fig. 9e). The relative difference of RMSD suggests that Exp.
**250m_8HARs** increases 40% accuracy of predicting characteristic half-widths by comparing to Exp. **500m_8HARs**. From the
perspective of computational resources, Exps. **250m_8HARs**, **500m_8HARs**, and **1000m_8HARs** spend $20.4 \times 10^4$ CPU
hours, $4.6 \times 10^4$ CPU hours, and $1.0 \times 10^4$ CPU hours, respectively.





In summary, the control run with a resolution of 500 m can basically reproduce the principal ISW field in the NSCS, while the sensitivity model with a higher resolution of 250 m would be a better solution to identify wave properties, in particular of the wave nonlinearity. Nonetheless, a 250 m-resolution model spends nearly fivefold computational resources of a 500 m-resolution model in the same model domain. Besides, the model with a lower resolution of 1000 m underestimates the nonlinearity of ISWs, thereby resulting in an inaccurate ISW field in the NSCS.

**4.2 Requirements of tidal constituents**

3D/2D models with different numbers of barotropic tidal constituents (e.g., single harmonic, four harmonics and eight harmonics) were commonly imposed to investigate the generation mechanisms of ISWs in the NSCS in previous studies (e.g., Li, 2014; Buijsman et al., 2010; Jin et al., 2021). However, whether a single tidal constituent can satisfy the reproduction of a real ISW field and how many tidal constituents are required for a realistic ISW model are still questions. Here, we run three sensitivity experiments (Exps. *500m_1HAR*, *500m_4HARs* and *500m_13HARs*) with different numbers of tidal harmonics to answer the questions by comparing the model performance with the control run (*500m_8HARs*).

We first discuss the model requirements of tidal constituents from the point of view of the ISW horizontal distributions and look back to Fig. 6. Note that time series of zonal barotropic currents at the generation site (Luzon Strait) are presented on the bottom left for each panel, where single/four/eight tidal constituent(s) are shown in green/magenta/blue. By comparing Exp. *500m_1HAR* (Fig. 6d) and *500m_8HARs* (Fig. 6a), we find that the single M2 tidal harmonic is not adequate to reproduce ISWs in the NSCS, so only some linear internal tides are detected on the sea surface via $|\nabla\eta|$. In contrast, Exp. *500m_4HARs* (Fig. 6e) nearly recreates the analogous scenario of ISWs to Exp. *500m_8HARs*, where IWB1, IWA1 and IWB2 appear at the same locations. Nonetheless, the crestline length (~134 km) of IWB2 in Exp. *500m_4HARs* is slightly shorter than that (~167 km) in Exp. *500m_8HARs*, and the secondary waves of IWB2 are unclear in Exp. *500m_4HARs* (see Fig. 6e). $|\nabla\eta|$ in Exp. *500m_13HARs* are not presented in Fig. 6, since it shows the exact same spatial patterns of ISWs as those in Exp. *500m_8HARs*, suggesting the principle eight tidal constituents are fine enough to satisfy accurate reproduction of the horizontal features of ISWs in a realistic oceanic model.

We then consider the difference of ISW vertical structures in sensitivity experiments with various tidal forcing via the selected transect and mooring station DS. In Exp. *500m_1HAR*, only linear internal waves are captured from the generation site to the slope, suggesting that single M2 tidal constituent without amplification factors can only contributes to internal tides and linear internal wave beams in NSCS (see Figs. 7d and 8e), unless the magnitudes of M2 barotropic tides are amplified, ISWs are likely to be generated (e.g., Yuan et al., 2020). In Exp. *500m_4HARs* (Figs. 7e and 8f), the single soliton IWA1 is reproduced with a smaller amplitude and weaker nonlinearity than that in Exp. *500m_8HARs*. Besides, the secondary waves of IWB2 are barely observed in Exp. *500m_4HARs*, which are much clearer in Exp. *500m_8HARs* (Figs. 7a and 8a). Figs. 8a and 8g depict the striking similarity of wave characteristics between Exp. *500m_8HARs* and Exp. *500m_13HARs*.

Last, we quantitatively estimate the sensitivity model capability of reproducing ISWs, by computing the biases and RMSDs of five wave properties (see Fig. 9 and Table 3) in the cases with different tidal forcing. Since Exp. *500m_1HAR* cannot predict





ISWs with significant amplitudes, we exclude it in the following analysis. In terms of Exp. ***500m_13HARs*** with thirteen tidal
constituents, the biases and RMSDs of five wave properties are very close to those in the control run with eight harmonics (see
overlapped black and cyan circles in Fig. 9 and Table 3). Conversely, Exp. ***500m_4HARs*** shows significant difference in the
biases and RMSDs of five wave properties from the control run. Specifically, in Fig. 9a, the RMSD of arrival time (0.81 h) is
larger in Exp. ***500m_4HARs*** than that in Exp. ***500m_8HARs*** (0.71 h). In addition, Exp. ***500m_4HARs*** underestimates averaged
wave-induced velocity for about 38% and averaged mode-1 wave amplitude for about 15%, which result in large negative
values of biases (see magenta circles in Figs. 9b and 9c), corresponding to 0.58 m s$^{-1}$ and 43.69 m of RMSDs, respectively. In
terms of the characteristic half-widths, Exps. ***500m_4HARs*** and ***500m_13HARs*** with RMSDs of 1.10 and 1.01 km show
analogous performance to the control run Exp. ***500m_8HARs*** with a RMSD of 1.07 km.

In summary, the model with eight (or thirteen) primary tidal constituents can accurately reproduce the real ISW field in the

NSCS, while the sensitivity model with four key harmonics (M2, S2, K1 and O1) would underestimate the magnitudes of
some secondary wave within a wave packet. In addition, the model only driven by M2 tide can only characterize wave
properties of linear internal waves (tides) instead of ISWs.

**4.3 Initial stratification selections**

As ISWs generate via tide-topography interaction in the stratified water, the stratification selection is crucial to directly

affect the model capabilities. Here, we extract the background stratification from the in-situ measurements at mooring station
DS as initial condition to run the sensitivity experiment ***500m_Real_N2***, and compare the model results with the control run
(***500m_8HARs***) with a climatological stratification from the WOA18 dataset.

In the model results, the spatial distribution of $|\nabla\eta|$ in Exp. ***500m_Real_N2*** shows analogous pattern of ISWs to that in Exp.

***500m_8HARs***. Specifically, three ISWs (i.e., IWB1, IWA1, and IWB2) appear at the same location in the two experiments
with similar horizontal wave characteristics (Figs. 6a and 6f). The visible difference is that the crest line length of the secondary
wave of IWB2 is longer with a stronger nonlinearity in Exp. ***500m_Real_N2***. We then look over the difference of ISW vertical
structures between two cases from the perspective of *x-z* plane along the transect (Figs. 7a and 7f) and time series at station
DS (Figs. 8a and 8h). It is clearly shown that Exp. ***500m_Real_N2*** with the real stratification can better characterize the
nonlinearity of the single soliton IWA1 and the secondary wave of wave train IWB2. Besides, the comparison with field
measurements reveals that Exp. ***500m_Real_N2*** shows a better precision (13%) in predicting the arrival time (i.e., RMSD of
0.62 h) of ISWs than the control run (i.e., RMSD of 0.71 h) with the climatological stratification. However, the RMSD of the
propagation direction of ISWs is larger in the realistic-stratification case (14.74°) than that of the control run (8.35°). Last, Exp.
***500m_Real_N2*** nicely describes the characteristic half-widths of ISWs (RMSD of 0.58 km), which improves 46% accuracy
by comparing that in Exp. ***500m_8HARs*** (RMSD of 1.07 km). To sum up, although the model with climatological stratification
works well, applying the real background stratification as the model initial condition would improve the model performance
in predicting some wave properties, including arrival time, wave-induced velocity, wave amplitude and characteristic half-
width.



## 5 Discussion and Conclusions

Although the three-dimensional realistic model, particularly in Exp. ***250m_8HARs***, has accurately reproduced the ISW features in the NSCS to some extent, the depictions of soliton numbers within an internal wave packet and propagation direction still have space for improvement, i.e., at least three following factors might be considered in the future modelling.

The first factor, that may affect the model accuracy, is background currents. Here, we download the HYCOM dataset in 2014 and calculate the background current field by averaging from 05-AUG to 20-AUG, namely predicting time of the model (see Fig. 10a). In Fig. 10a, there is a clear counter-clockwise circulation/eddy pattern on the west side of Luzon Strait. Xie et al. (2015) suggested that wave properties of ISWs can be significantly influenced by an isolated mesoscale eddy, regardless of a cyclonic or anticyclonic eddy, during the propagation of ISWs. When an ISW pass over a cyclonic eddy, as in Fig. 10a, the crestline will be distorted, thereby modulating the oblique propagation direction of wave to some extent (Xie et al., 2016). In addition, a series of secondary trailing waves are able to form behind the leading wave in the energy-focusing region. Therefore, background currents are supposed to be considered in the future forecasting model, which shows potential improvement in the depiction of soliton numbers within an ISW packet and propagation direction in the NSCS.

The second factor is inhomogeneous spatial distribution of stratification. In the current forecasting model, we apply horizontally-homogeneous temperature and salinity profiles (Fig. 1d) with the maximum buoyancy frequency of ~0.02 $s^{-1}$ at a water depth of 50 m. However, the stratification is spatially varying in the real ocean (see Fig. 10b, time-averaged buoyancy frequency derived from the HYCOM dataset), although buoyancy frequency is ranging from 0.015 to 0.025 $s^{-1}$ in the most model region. Since wave speeds of ISWs and internal tides are closely related to vertical structure of stratification based on eigen-function, the inhomogeneous stratification pattern is likely to affect ISW propagation speed and then modulate their arrival time. Most of previous numerical studies (e.g., Zhang et al., 2011; Alford et al., 2015; Zeng et al., 2019) rarely considered the impacts of horizontally inhomogeneous stratification, but Chi et al. (2016) and Lai et al. (2019) applied spatially varying stratification in 3D model and indicated that inhomogeneous stratification can achieve better model results to some extent. Hence, spatially varying stratification is worthwhile to be considered in future numerical studies in the NSCS.

The last element is external (wind) forcing. As is well known, internal waves are a ubiquitous phenomenon, of which maximum amplitudes happen in the ocean interior. Nonetheless, the thermoclines usually occur in the upper layers (shallower than 500 m) in the SCS, which can be significantly affected by extreme wind events (i.e., tropical cyclones, Zhang 2022). So far, wind forcing was rarely applied in the numerical modelling of ISWs, except Lai et al. (2019). As both the ISWs and tropical cyclones are active and frequent in August, September and October in the SCS, the impacts of tropical cyclones on the upper layers should be considered in the future numerical simulations, although tropical cyclone does not happen during our predicting period (see Fig. 10c).

In summary, this study introduces a robust ISW forecasting model by comparing with in-situ observational data and remote-sensing images, and quantitatively evaluates the requirements of different factors, including the horizontal resolutions, tidal





constituents and initial stratification, for accurately characterizing the ISW field with applications to the NSCS. The major

findings are listed as follows.

1)    A model with a 500 m resolution can basically reproduce the principal ISW field, while a model with a higher resolution
      of 250 m would be a better solution to identify wave properties but spends nearly fivefold computational resources of
      a 500 m-resolution case with the same model domain.

2)    At least eight primary tidal constituents should be included in the boundary forcing.

3)    Compared to climatological stratification, applying the observational background stratification could improve the
      model performance in predicting some wave properties, namely 13% improvement of arrival time and 46%
      improvement of characteristic half-width.

**Appendix A: Feasibility study of two-dimensional slice model**

Differing from the 3D models, 2D slice models are fairly economical from the perspective of computational resources. In
the past few decades, 2D slice models with idealized topography (double ridges) were commonly conducted to investigate
ISW dynamics in the NSCS, in particular for the generation mechanisms and the affecting factors of ISWs (i.e., Cai et al.,
2002; Shaw et al., 2009; Li, 2014). Here, we attempt to test the 2D model performance along different transects and clarify
whether a 2D slice model can be a substitute for a 3D model in the aspect of reproducing a real ISW field in the NSCS.
Three parallel transects with a distance of 0.05° are selected along the main propagation direction of ISWs (see dashed lines
in Fig. 1a), which are labelled as **2D_500m_8HARs**, **2D_500m_8HARs_005N**, and **2D_500m_8HARs_005S**. $\Delta x$ and $\Delta t$ are
still set as 500 m and 10 sec, respectively. Initial conditions and dissipation coefficients are set the same as those in the 3D
control run (**500m_8HARs**). The 2D slice models are also driven by the barotropic tides of eight tidal constituents at both west
boundary (115.8°E, 21.1°N $\pm$ 0.05°) and east boundary (123.8°E, 19.5°N $\pm$ 0.05°). As the transects are not strictly zonal (angle
$\theta$ = 11.2°, see Fig. 1a), it is necessary to extract the amplitude ($U'$) and phase ($\phi'$) for each harmonic ($\omega$) in the transect
direction from the TPXO8-atlas dataset (i.e., $U$, $V$, $\phi_U$, and $\phi_V$), given by

$$U' = \sqrt{(U \cdot cos\phi_U cos\theta - V \cdot cos\phi_V sin\theta)^2 + (U \cdot sin\phi_U cos\theta - V \cdot sin\phi_V sin\theta)^2}, \qquad (A1)$$

$$\phi' = arctan\left(\frac{U \cdot sin\phi_U cos\theta - V \cdot sin\phi_V sin\theta}{U \cdot cos\phi_U cos\theta - V \cdot cos\phi_V sin\theta}\right), \qquad (A2)$$

Here, we apply the standard 2D experiment along the selected transect (see the black dashed line in Fig. 1a) and label it as
**2D_500m_8HARs**. The model is driven by eight principle tidal constituents on the both lateral boundaries, which are extracted
from the TPXO8 dataset (following Eqns. A1 and A2). Note that initial conditions and other model configurations in Exp.
**2D_500m_8HARs** are the same as those in the 3D control run (**500m_8HARs**). In addition, we run two sensitivity experiments
(Exps. **2D_500m_8HARs_005N** and **2D_500m_8HARs_005S**) along the two parallel transects (see red dashed lines in Fig.
1b).





In the 2D standard case (*2D_500m_8HARs*), ISWs subsequently generate in the double ridge, then propagate westward, and eventually arrive at the station in the form of wave trains (Fig. A1b). The wave amplitudes are greater than those in the 3D control run (Fig. A1a). At the station outputs (Fig. A1f), we find that Exp. *2D_500m_8HARs* can only reproduce ISW packets, but cannot discriminate type-a and type-b ISWs. Although the occurrence frequency of ISWs is also twice per day in Exp. *2D_500m_8HARs*, the arrival time of those ISW packets is not consistent with that in Exp. *500m_8HARs* (Fig. A1e) and in the field measurements (Fig. 8a). In Exp. *2D_500m_8HARs_005N*, ISWs are rarely found along the transect (Fig. A1c), likely due to the relatively gentle topography and small tidal forcing at the lateral boundaries. At the station outputs (Fig. A1g), only small temperature fluctuations are captured. Conversely, Exp. *2D_500m_8HARs_005S* show analogous wave fields to Exp. *2D_500m_8HARs* (Fig. A1d). Specifically, ISW packets with a half-day cycle are dominant, but their arrival time are postponed for about two hours (Fig. A1h).

To sum up, 2D slice models along different transects (even 0.05º apart) present totally different ISW characteristics, which are inconsistent with the 3D model results and in-situ measurements. Therefore, 3D model is the best and sole option to correctly reproduce the ISW field in the real ocean, while 2D model is more suitable for the mechanism investigations.

*Code and data availability*. The MODIS remote-sensing images are derived from the NASA Worldview application (https://worldview.earthdata.nasa.gov). The Massachusetts Institute of Technology general circulation model used for simulating nonlinear internal waves is available at https://mitgcm.org/source-code/ and developed openly at https://github.com/MITgcm/MITgcm/commits/master. The input files (including initial and boundary conditions) and relevant output data files of the three-dimensional realistic model in the northern South China Sea are available at a free, open access, data repository via https://doi.org/10.5281/zenodo.6792999. The field observational data at the Dongsha mooring station is available at the repository via https://doi.org/10.5281/zenodo.6793125.

*Author contributions*. YG wrote the paper with the help of all the co-authors. XC, JX, JX, ZC, YH and SC provided constructive feedback on the manuscript. JX gave help and advice in observational data processing and numerical simulations.

*Competing interests*. The authors declare that they have no conflict of interest.

*Financial support*. This work was jointly supported by the National Natural Science Foundation of China (NSFC) under contract Nos. 42130404, 91858201, 42206012, 42276015, 42276022 and 42176025; the Key Research Program of Frontier Sciences, Chinese Academy of Sciences (CAS) under contract No. QYZDJ-SSW-DQC034; the China Postdoctoral Science Foundation (2022M713232); Grant No. ISEE2021PY01 from CAS; Youth science and technology innovation talent of Guangdong TeZhi plan (2019TQ05H519); Rising Star Foundation of SCSIO (NHXX2019WL0201); Natural Science Foundation of Guangdong Province (2020A1515010495, 2021A1515012538, 2021A1515011613); the Youth Innovation Promotion Association from CAS (2019336); the State Key Laboratory of Tropical Oceanography Independent Research



Program under contract No. LTOZZ2205. The numerical simulation is supported by the High Performance Computing
Division and HPC managers of Wei Zhou and Dandan Sui in the South China Sea Institute of Oceanology.

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







**Figure 1.** (a) Bathymetry map of model domain in the northern South China Sea with a mooring station DS (marked as magenta star) in the vicinity of Dongsha Atoll and the transects in two-dimensional models, among which Exp. *2D_500m_8HARs* is in black dashed line while Exps. *2D_500m_8HARs_005N* and *2D_500m_8HARs_005S* are in red dashed lines. (b) Initial temperature and salinity profiles. (c) Density profile. (d) Buoyancy frequency profile. Note the black and red lines in (b-d) represent the data derived from the WOA18 and in-situ observations, respectively.





595

**Figure 2.** Absolute root-mean-square errors of zonal barotropic velocity ($U_{bt}$) between the model (500m_8HARs_BT) and the TPXO8-Atlas dataset for M2 (a), S2 (b), K1 (c), and O1 (d). (e) Reconstructed time series of zonal barotropic velocity at station DS (marked as magenta star in Fig. 2a) of Exp. **500m_8HARs_BT** (black line) versus measured data (red line) obtained by eight key tidal constituents.



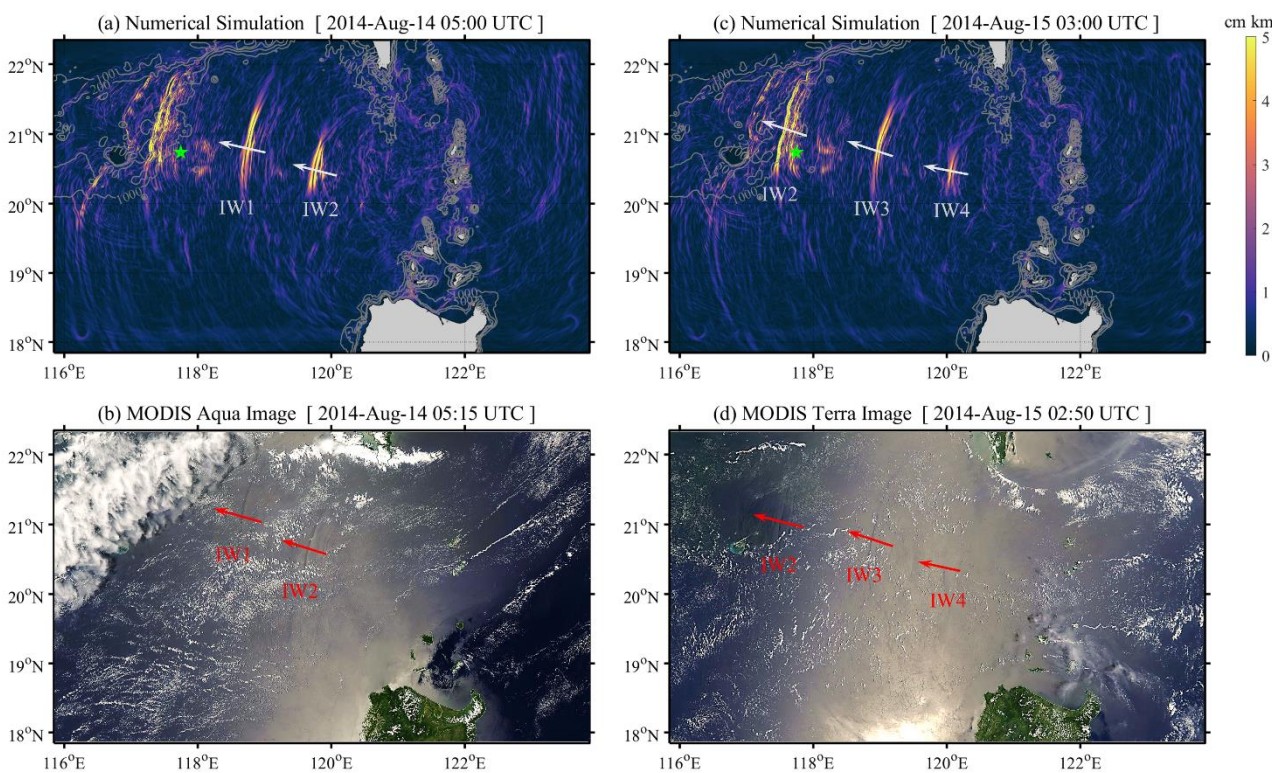

600

**Figure 3.** (a) Sea surface height gradients induced by internal solitary waves (ISWs) at 05:00 UTC on 14 August 2014 and (b) MODIS-Aqua image obtained at 05:15 UTC on 14 August 2014. (c) Same as (a) but at 03:00 UTC on 15 August 2014. (d) Same as (b) but for MODIS-Terra at 02:50 UTC on 15 August 2014. Note that the MODIS images in (b) and (d) are freely downloaded from the NASA Worldview application (https://worldview.earthdata.nasa.gov, open source).

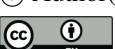

605

**Figure 4.** (a - g) Temperature isotherms (contours) and baroclinic velocities (shades) in the wave propagation direction from 08 August to 14 August at station DS from in-situ observation. (h – n) Same as (a – g) but for the model (**500m_8HARs**). Red arrows indicate ISWs that model captured, while blue arrows present the missed ones.



**Figure 5.** Maximum wave-induced velocities (a), propagation directions (b) and maximum mode-1 wave amplitudes (c) of fifteen ISWs at station DS from in-situ observations (red) and numerical models (green). Averaged values are shown by solid lines.





**Figure 6.** Sea surface height gradients at 12:00 UTC on 12 August 2014 in the model (a) *500m_8HARs*, (b) *250m_8HARs*, (c) *1000m_8HARs*, (d) *500m_1HAR*, (e) *500m_4HARs*, and (f) *500m_Real_N2*. Note that dashed line in (a) is selected transect to present vertical structure of ISWs. Small panels on the bottom left indicate the zonal barotropic velocity (unit in m s$^{-1}$) in the Luzon Strait with the solid lines showing the tidal conditions at the selected time.

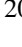

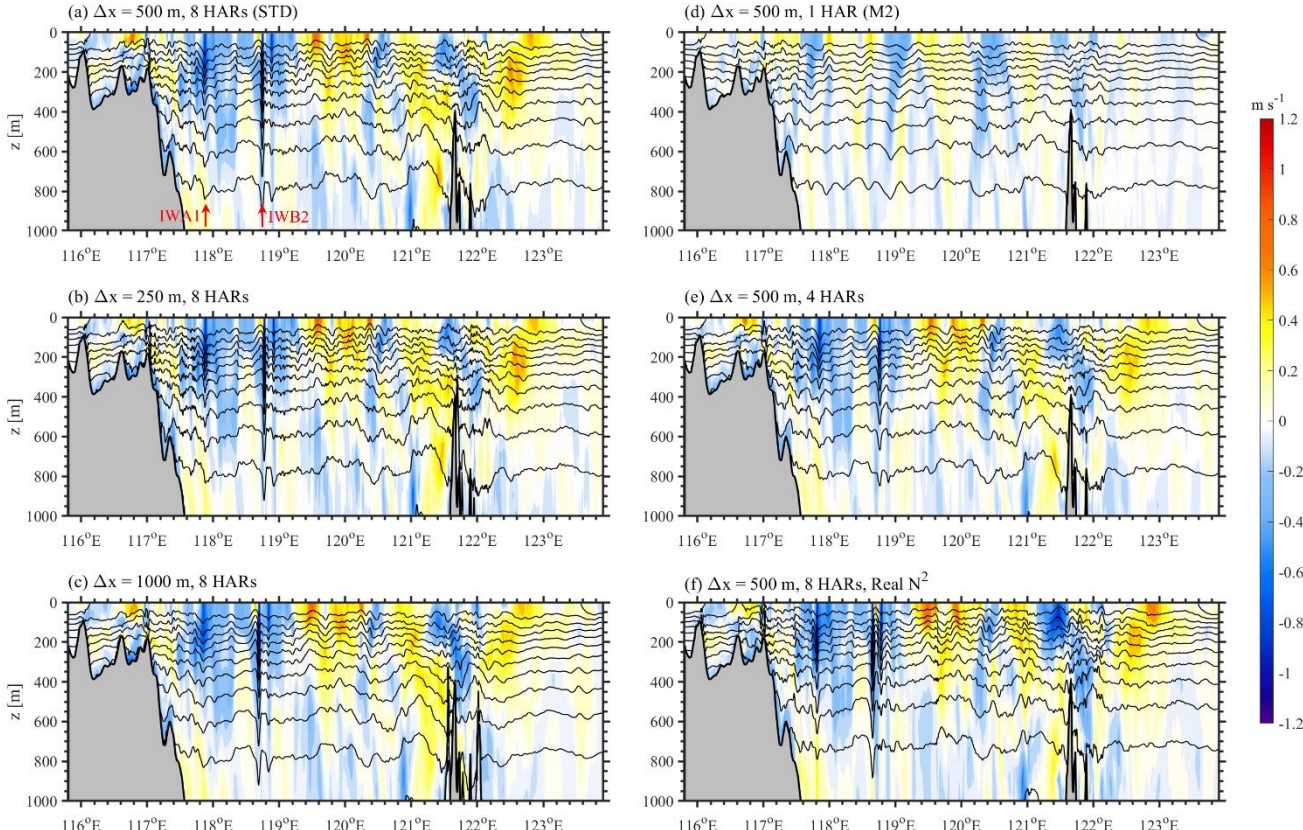

**Figure 7.** Temperature isotherms (contours) and baroclinic velocities (shades) along the transect (dashed line in Fig. 6a) at
12:00 UTC on 12 August 2014 in the model (a) ***500m_8HARs***, (b) ***250m_8HARs***, (c) ***1000m_8HARs***, (d) ***500m_1HAR***, (e)
***500m_4HARs***, and (f) ***500m_Real_N2***. Note that waves IWA1 and IWB2 are labelled in (a) with red arrows.



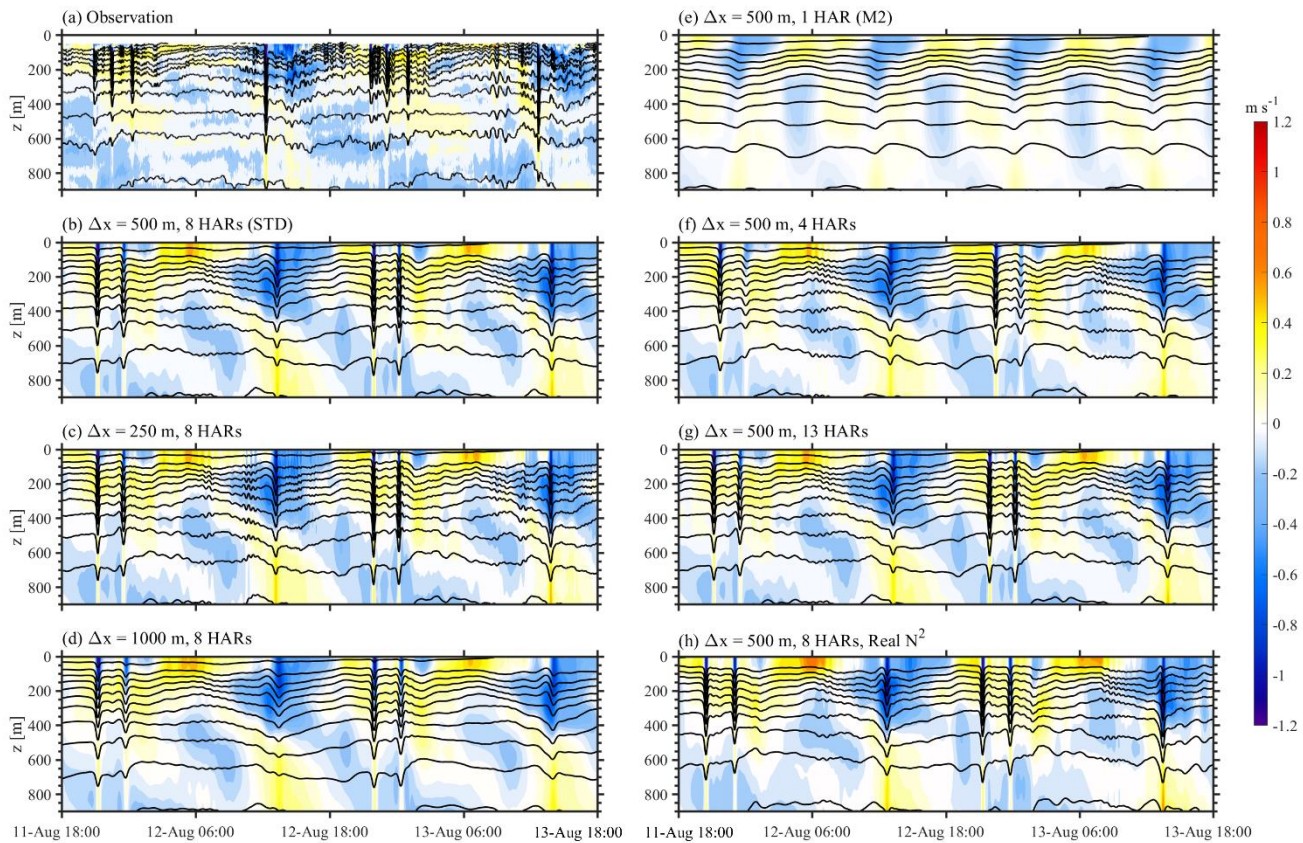

**Figure 8.** Time series of temperature isotherms (contours) and baroclinic velocities (shades) at station DS from 18:00 UTC on 11 August to 18:00 UTC on 13 August 2014 in the observation (a) and in the model (b) *500m_8HARs*, (c) *250m_8HARs*, (d) 625 *1000m_8HARs*, (e) *500m_1HAR*, (f) *500m_4HARs*, (g) *500m_13HARs*, and (h) *500m_Real_N2*.



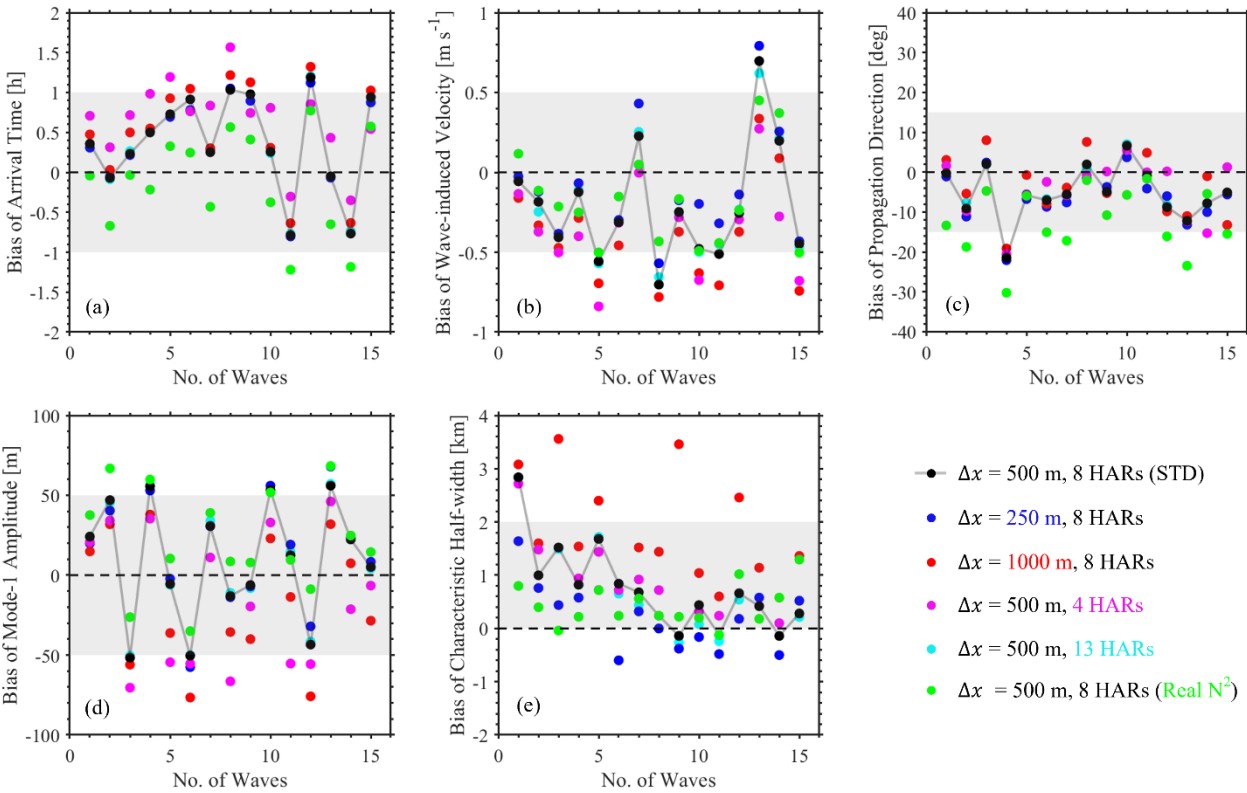

**Figure 9.** Bias of arrival time (a), maximum wave-induced velocities (b), propagation directions (c), maximum mode-1 wave amplitudes (d), and characteristic half-widths (e) for fifteen ISWs at station DS. Colours present different experiments.



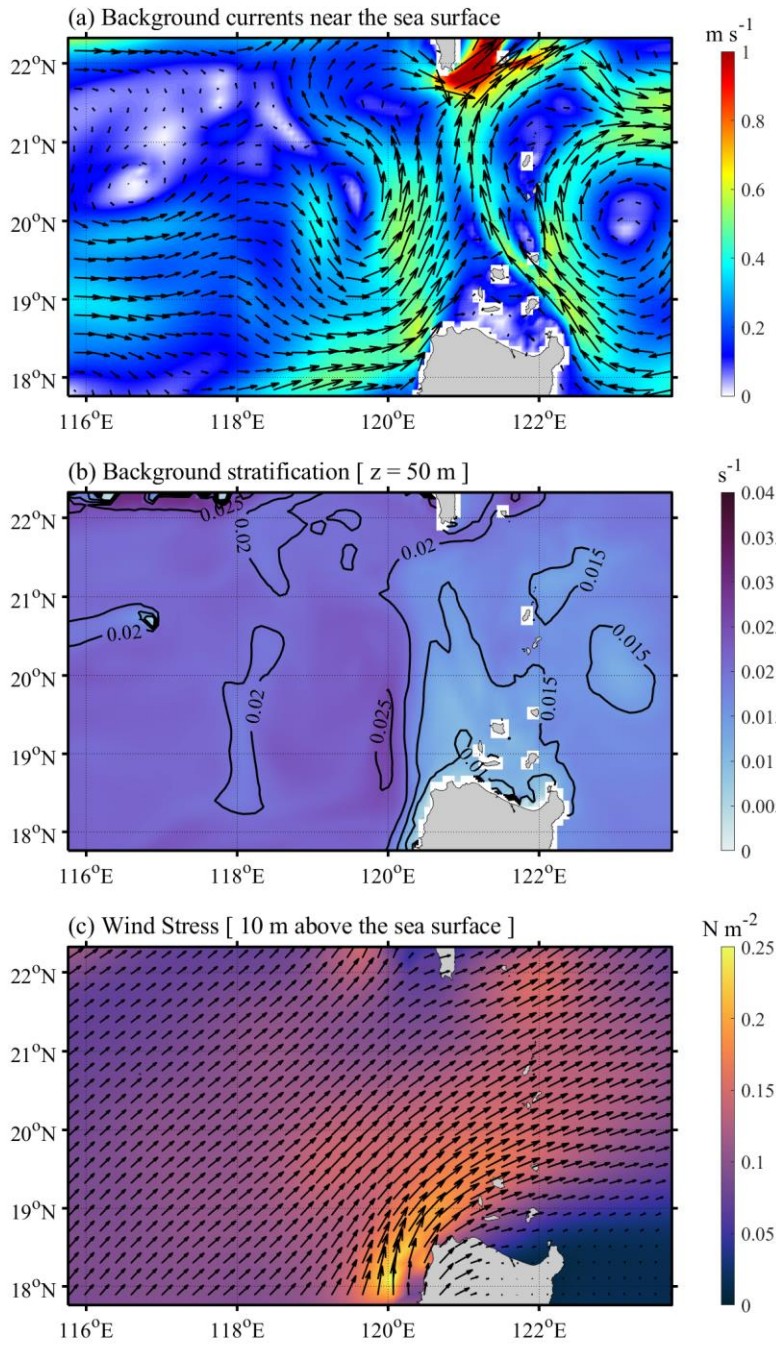

**Figure 10.** (a) Background currents near the sea surface (averaged from 05-AUG to 20-AUG 2014, derived from HYCOM dataset). (b) Background buoyancy frequency at a water depth of 50 m. (c) Time-averaged wind stress at 10 m above the sea surface, which is derived from NCEPv2 hourly dataset.



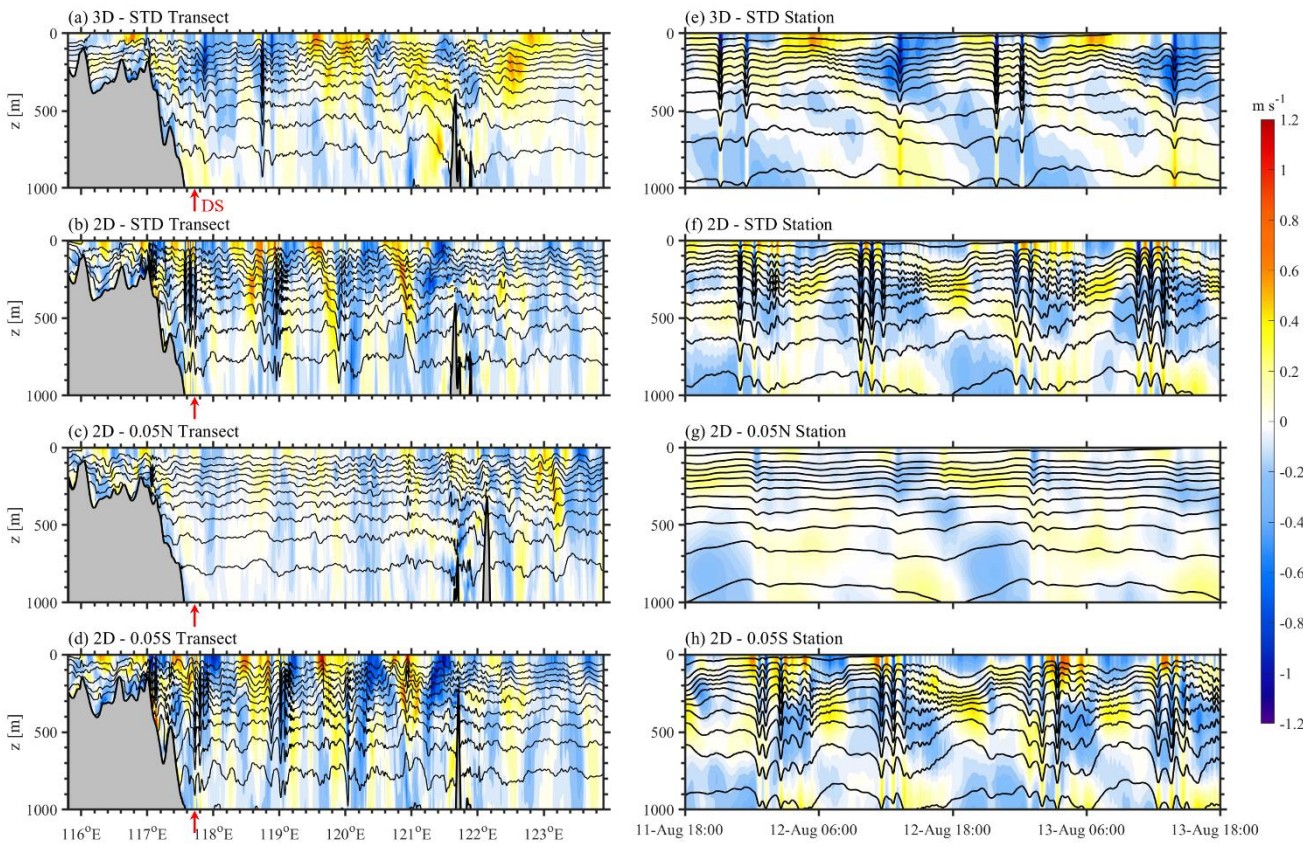

**Figure A1.** Temperature isotherms (contours) and baroclinic velocities (shades) along the transect at 12:00 UTC on 12 August 2014 in the three-dimensional model (a) *500m_8HARs*, in the two-dimensional model (b) *2D_500m_8HARs*, (c) *2D_500m_8HARs_005N*, and (d) *2D_500m_8HARs_005S*. (e – h) Corresponding time series at the stations, which are marked as red arrows in (a – d).





**Table 1.** Summary of previous three-dimensional non-hydrostatic models for internal solitary waves in the northern South China Sea, which are discussed in the text. Further details can be found in the references.

| References | Model | Resolution | Tidal constituents | Model domain |
|---|---|---|---|---|
| Vlasenko et al. (2010) <br> Guo et al. (2011) | MITgcm | $\Delta x$ = 250 m, $\Delta y$ = 1000 m | 8 HARs | 118.0º – 122.5ºE <br> 20.0º – 21.0ºN |
| Zhang et al. (2011) | SUNTANS | ~1358 m (75 – 4740 m) | 8 HARs | 115.0º – 124.0ºE <br> 18.0º – 23.0ºN |
| Alford et al. (2015) | MITgcm | 250 m | 8 HARs | 119.6º – 122.3ºE <br> 18.8º – 21.8ºN |
| Lai et al. (2019) | FVCOM | ~200 – 500 m (near the shoreline) <br> ~3 km (shelf-slope region) | 8 HARs | 105.0º – 130.0ºE <br> 12.0º – 30.0ºN |
| Zeng et al. (2019) | MITgcm | $\Delta x$ = 150 m, $\Delta y$ = 300 m | 8 HARs | 115.5º – 124.5ºE <br> 17.5º – 22.5ºN |





**Table 2.** Summary of all experimental configurations.

| No. | Experiment name | Grid spacing | Tidal forcing | Stratification |
|:---:|:---:|:---:|:---:|:---:|
| **1** | *500m_8HARs* | 500 m | 8 HARs (M2, S2, N2, K2, K1, O1, P1, Q1) | WOA18 |
| **2** | *500m_8HARs_BT* | 500 m | 8 HARs | - |
| **3** | *250m_8HARs* | 250 m | 8 HARs | WOA18 |
| **4** | *1000m_8HARs* | 1000 m | 8 HARs | WOA18 |
| **5** | *500m_1HAR* | 500 m | 1 HAR (M2) | WOA18 |
| **6** | *500m_4HARs* | 500 m | 4 HARs (M2, S2, K1, O1) | WOA18 |
| **7** | *500m_13HARs* | 500 m | 13 HARs (M2, S2, N2, K2, K1, O1, P1, Q1, M4, MS4, MN4, MM, MF) | WOA18 |
| **8** | *500m_Real_N2* | 500 m | 8 HARs | DS Station |

645





**Table 3.** Root mean square deviation (RMSD) of wave properties between field observation and 3D sensitivity simulations at the mooring station in the vicinity of the Dongsha Atoll.

| No. | Experiment name | RMSD of arrival time [h] | RMSD of wave-induced velocity [m s⁻¹] | RMSD of propagation direction [º] | RMSD of mode-1 wave Amplitude [m] | RMSD of characteristic half-width [km] |
|---|---|---|---|---|---|---|
| 1 | *500m_8HARs* | 0.71 | 0.41 | 8.35 | 37.27 | 1.07 |
| 2 | *500m_8HARs_BT* | - | - | - | - | - |
| 3 | *250m_8HARs* | 0.67 | 0.38 | 8.89 | 38.12 | 0.64 |
| 4 | *1000m_8HARs* | 0.79 | 0.49 | 8.54 | 40.28 | 2.41 |
| 5 | *500m_1HAR* | - | - | - | - | - |
| 6 | *500m_4HARs* | 0.81 | 0.58 | 8.22 | 43.69 | 1.10 |
| 7 | *500m_13HARs* | 0.71 | 0.40 | 8.23 | 37.36 | 1.01 |
| 8 | *500m_Real_N2* | 0.62 | 0.34 | 14.74 | 37.88 | 0.58 |