# Peer review of "An internal solitary wave forecasting model in the northern South China Sea (ISWFM-NSCS)"

_Geoscientific Model Development, 2022_

## Author Comment (AC1)

**Response letter to the referee #1**

Review of "An internal solitary wave forecasting model in the northern South China Sea (ISWFM-NSCS)" by Gong et al.

Reported is a three-dimensional internal solitary wave forecasting model in the northern South China Sea. The model performance was quantitatively evaluated with mooring observations. A series numerical experiments were implemented to test the choice of horizontal resolution, the minimum number of tidal constituents, as well as background stratification. The work is interesting and the results contribute to the advances of the ISWs modeling in the northern South China Sea. I would like to recommend the manuscript be accepted for publication after very minor technical corrections that are listed as follows:

*Response*:

We would like to thank the referee for this careful reading and valuable comments. In the revision, we have carefully considered them, and the necessary changes are provided to address them. Below, we provided point-by-point responses in blue to your comments.

In this manuscript, "three-dimensional" and "3D" are used interchangeably. Better to make them consistent throughout the text.

*Response*: We now mention the abbreviation of "three-dimensional" in the Introduction section, and use "3D" throughout the following text to keep the consistency.

Line 17, "precisely" should be changed to "accurately".

*Response*: We now replace "precisely" with "accurately".

Lines 59-61, "The questions that arise are whether a single tidal constituent can satisfy the reproduction of a real ISW field and how many tidal constituents are required for running an accurate 3D realistic ISW model." I would suggest to rewrite the sentence as "The questions are whether a single tidal constituent can reproduce a real ISW field and, if not, how many tidal constituents are required for running an accurate 3D realistic ISW model."

*Response*: We now revise this sentence to "The questions are whether a single tidal constituent can reproduce a real ISW field and, if not, how many tidal constituents are required for running an accurate 3D realistic ISW model."

Lines 105-106, "The control run (500m_8HARs) runs at 1 h time interval" is confusing. Did you mean model output at one hour interval?

***Response***: We agree the description of sampling rate is a bit confusing and now we revise it to "The sampling rate of model outputs is 1 h interval for the entire model domain in the control run (***500m_8HARs***)".

Lines 126-127, "respectively" should be added to the end of the sentences.

***Response***: We now add "respectively" at the end of the sentences.

Line 204, "them" should be removed from the sentence.

***Response***: We now remove "them" in this sentence.

Line 204, "In terms of wave propagation direction, we obtain by computing the angle …" should be written as "We obtain wave propagation direction by computing the angle …"

***Response***: We now revise this sentence to "We obtain wave propagation direction by computing the angle of baroclinic zonal and meridional components in the layer with maximum velocity".

Line 382, "HYCOM" should be spelled out in the first time use in the paper.   Also, it should be global HYCOM.

***Response***: We now spell out "HYCOM" as "Hybrid Coordinate Ocean Model" and add "global" in the front.

Lines 401-402, "… phenomenon, of which maximum amplitudes happen in the ocean interior" should be written as "… phenomenon, with maximum amplitudes in the ocean interior".

***Response***: We now revise this sentence to "internal waves are a ubiquitous phenomenon with maximum amplitudes in the ocean interior".

Line 406, "does" should be changed to "did".

***Response***: "Does" has been replaced with "did".

---

## Author Comment (AC2)

**Response letter to the Editor**

You have archived your code (the MITgcm model) on GitHub and MIT servers. However, these are not acceptable repositories. You must store the code used in your manuscript in other long-term archival alternatives, such as Zenodo, PANGAEA, etc.

Therefore, you must reply to this comment with the relevant information (link and DOI) for the new repositories, as we request that you make it already available before submission and, of course, before the Discussions stage.

Moreover, you must include in a potentially revised version of your manuscript the modified 'Code and Data Availability' section, with the DOI of the code.

*Response*:

We appreciate your comments on our code repository. According to your suggestion, we have read the code and data policy of GMD in detail, and have added all model code on the Zenodo repository with the DOI number https://doi.org/10.5281/zenodo.6792999.

Therefore, the "Code and data availability" is now rewritten as "The MODIS remote-sensing images are derived from the NASA Worldview application (https://worldview.earthdata.nasa.gov). The input files (including initial and boundary conditions) and relevant output data files of the 3D realistic Massachusetts Institute of Technology general circulation model in the northern South China Sea are available at a free, open access, data repository via https://doi.org/10.5281/zenodo.6792999".

---

## Author Comment (AC3)

**Response letter to the referee #2**

The article simulated the propagation of internal solitary waves in the northern South China Sea using a three-dimensional ISW forecasting model. The results are compared with satellite in situ mooring observations, showing good consistency. A series of sensitivity experiments are conducted to further study effects of model resolution, tidal forcing, and stratification selection on the model simulations. This article is very well written, and the detailed analysis in this article show robust conclusions. This has the potential to advance our understanding of developing an internal solitary waves real-time forecasting system. Therefore, I recommend this article to be published in Geoscientific Model Development with very minor revisions. My suggestions are listed below.

***Response*:**

We would like to thank the referee for this careful reading and valuable comments. In the revision, we have carefully considered them, and the necessary changes are provided to address them. Below, we provided point-by-point responses in blue to your comments.

Lines 89-91 and 391-400: The authors used the spatially averaged temperature and salinity profiles from WOA18 as the initial condition for the 3D model simulation. In the section of 'discussion and conclusions', the authors discussed that the spatially varying stratification needs to be consider in the future work. Using the spatially varying stratification is easy to be implemented and will not increase the computational resources. I wonder if this can be included in this work. In addition, WOA18 is a climatology dataset, a three-dimensional temperature and salinity profiles from high resolution ocean reanalysis, such as HYCOM, GLORYS and BRAN2020, may give a better performance of the control run.

***Response*:**

We appreciate the referee pointing out that the spatially varying stratification should be implemented in the model. Actually, we have applied spatially varying initial conditions with WOA18 climatology dataset and run a sensitivity experiment (3D-TS). By comparing the sensitivity experiment (3D-TS) and the control run (***800m_HARs***), it is concluded that the spatially varying initial conditions show similar performance in predicting the arrival time and horizontal distributions (Figure R1), wave amplitudes and wave-induced velocities (Figure R2) of ISWs to the horizontally homogeneous initial conditions. Therefore, we remain the horizontally homogeneous stratification as the model initial conditions in this work.

As for high-resolution ocean reanalysis dataset (e.g., HYCOM, GLORYS and BRAN2020), we admit that the model results would be different with spatially varying stratification from these datasets.

However, only considering spatially varying temperature and salinity profiles with large values of horizontal gradients (e.g., HYCOM dataset in Figure 10b) as the initial conditions might lead to spurious geostrophic currents, thereby significantly affecting the true wave field.

Therefore, as we discuss in the paper, we'd prefer to consider both horizontally inhomogeneous stratification profiles and background currents in future studies, rather than only consider horizontally inhomogeneous stratification profiles.

[Figure]

Figure R1. (a-c) Snapshots of sea surface gradients ($|\nabla\eta|$) at 12:00 UTC on 11, 12 and 13 August in the standard experiment (EXP. ***500m_8HARs***), respectively. (d-f) Same as (a-c) but in the sensitivity experiment with horizontally inhomogeneous temperature and salinity (3D-TS).

[Figure]

Figure R2. (a-c) Temperature isotherms (contours) and baroclinic velocities (shades) along the transect (dashed line in Figure R1a) at 12:00 UTC on 11, 12 and 13 August in the standard experiment (EXP. *500m_8HARs*), respectively. (d-f) Same as (a-c) but in the sensitivity experiment with horizontally inhomogeneous temperature and salinity (3D-TS).

Lines 96-97: Why not use spatially varying Coriolis parameters? Are the model results sensitive to this parameter?

*Response*: We appreciate the reviewer pointing this out. Actually, we did implement all standard and sensitivity experiments with latitudinally varying Coriolis coefficients, but made a mistake in the manuscript. Please check our model configuration file "data" in the Zenodo link, in which "selectCoriMap = 2" indicates "Spherical Coriolis" (i.e., $f$ is varying with latitude). The mistake has now been fixed up in the revised manuscript.

Lines 104-105: As the model is initiated from the spatially averaged temperature and salinity profiles, is the model integrated time (3-day) enough for the 3D ocean model to reach a steady state?

***Response***: Due to the limitation of the computational resources, we admit that the model integrated time (3-day) is short. But when we calculate the averaged baroclinic kinetic energy ($KE_{bc}$) in the inner model domain (Figure R3), it is found that the depth-integrated $KE_{bc}$ keeps increasing in the first 7 days (see red line in Figure R3a), which is related to the flooding barotropic tides (see Figure R3b). Actually, $\int_{-H}^{0} KE_{bc}\, dz$ reaches 10 kJ m$^{-2}$ in the 3$^{rd}$ model day, whose magnitude is equivalent to that during the neap tides (i.e., 12$^{th}$ to 15$^{th}$ model day). It shows that the 3D model reaches a quasi-steady state after three days. The comparison between the model results and field observations at the mooring station DS verifies the model accuracy since 08 August (the 3$^{rd}$ model day) as well. Therefore, we consider 3 days as the spin-up time in this work.

[Figure]

Figure R3. (a) Domain-averaged baroclinic kinetic energy ($KE_{bc}$, in the unit of J m$^{-3}$) in the inner model domain. Note that the red solid line is domain-averaged depth-integrated $KE_{bc}$ ($\int_{-H}^{0} KE_{bc}\, dz$, in the unit of kJ m$^{-2}$). (b) Time series of barotropic velocity in the Luzon Strait.

Line 126: Because the mooring will fluctuate in the water, I do not think the CTD sensors can be fixed at 1 m which is too close to the sea surface.

*Response*: We agree that the CTD sensor at 1 m is too close to the sea surface, which is inappropriate to observe ISWs. Therefore, we remove this CTD sensor and remain the lower one at 1100 m.

Lines 173-175: How do you calculate the crestline lengths. Are there several biases for your estimation?

*Response*: As for the MODIS images, we first enhance the contrast between the NLIWs and the surrounding water via image processing software, then the crestlines of NLIWs would be clearly identified and easily extracted. As for the model results, we focus on the sea surface gradients ($|\nabla \eta|$) larger than 2 cm km$^{-1}$ along the crestlines and define the lengths as crestline lengths of NLIWs. We have now made the statement in the revised manuscript. We agree that this estimation method might cause different biases with different criterions, but $|\nabla \eta| > 2$ cm km$^{-1}$ seems to be a suitable criterion for evaluating NLIW crestline lengths by comparing to the MODIS images.

Lines 176-177: 'As the model ...... between them'. Will these factors impact the ISW parameters that calculated from the MODIS images?

*Response*: Yes, these factors have impacts on the ISW parameters calculated from the MODIS images. On the one hand, wind and clouds above the sea surface might affect the statistical analysis of ISW occurrence frequencies and evaluation of wave crestline lengths. On the other hand, background currents induced by marine dynamical processes would modulate the nonlinear and dispersion parameters of ISWs, thereby affecting the estimation accuracy of ISW characteristics (e.g., wave amplitudes).

Lines 184-186: Can you estimate the exact propagation speed of the ISWs occurred during 8 August-14 August? If so, you can get more accurate time for the ISWs to propagate from the generation site to the targeted station.

*Response*: Yes, we can estimate the exact averaged propagation speed based on the space-time diagram of isopycnal displacements (i.e., mode-1 amplitudes) along the main propagation path of ISWs. Here, we plot a space-time diagram of isopycnal displacements along the selected transect (see Figure R4) and point out the generating time of ISWs in the Luzon Strait and the arrival time at the target station, respectively. It is found that ISWs spend 1.5 days arriving at the station with a time-averaged propagation speed of ~3.0 m s$^{-1}$.

[Figure]

Figure R4. Space-time diagram of isopycnal displacements along the selected transect. Black vertical dashed line points out the location of generation site (~121.5 °E) and the targeted station is located at 117.75 °E. Black solid line shows the zonal barotropic velocity in the Luzon Strait.

Lines 354-357: Do these eight tidal constituents play the same role in the model simulation? I wonder whether we need all these eight tidal constituents?

**Response:** We agree that eight tidal constituents play different roles in generating NLIWs. Normally, M2, S2, K1 and O1 have been considered as the most important tidal constituents in the Luzon Strait (e.g., Fang et al., 1999; Wang et al., 2016). However, in this work, the experiment with only four tidal constituents (M2, S2, K1 and O1 in EXP. **500m_4HARs**) is not able to accurately reproduce NLIW

properties (see Table 3) by comparing the control run (EXP. *500m_8HARs*) with eight constituents, thereby confirming the importance of the residual four tidal constituents (N2, K2, P1, Q1). Overall, it is concluded that considering all these eight tidal constituents are necessary to improve the predicting accuracy of ISWs in the South China Sea.

Lines 412-414: The authors chose three different horizontal model resolutions (250 m, 500 m and 1000 m) and compared the model simulations. I wonder if the higher the horizontal model resolution is, the better the model will perform. What about the vertical model resolutions?

*Response*: By comparing the field observations with EXPs. *250m_8HARs*, *500m_8HARs*, and *1000m_8HARs* at the mooring station DS, we list the root mean square deviation (RMSD) of wave properties in Table 3. It is found that compared with EXP. *1000m_8HARs*, the model with a 500 m resolution (EXP. *500m_8HARs*) significantly improves the predicting accuracy of wave properties. However, the EXP. *250m_8HARs* only shows slight improvement of predicting performance in comparison to the EXP. *500m_8HARs*, but costs much more computational resources. Considering both model performance and time/CPU consumption, the EXP. *500m_8HARs* might be a better choice to accurately simulate ISW properties in the northern SCS.

Yes, we have not yet discussed the roles of vertical resolution in model performance. In this study, we apply 90 vertical layers in accordance with the hyperbolic tangent function (Stewart et al., 2017), ranging from 5 m near the surface to 120 m near the sea bed (in the deep water). Specifically, the vertical cell size is adequately high in the thermocline to accurately reproduce vertical characteristics of ISWs in the upper layers. We agree that vertical model resolutions could be a factor affecting the simulation accuracy, so we'd like to discuss the selection of vertical resolutions in future studies.

Figures 6-9: In Figs. 6f, 7f, 8h and 9, the labels should be Real N2 instead of Real $N^2$.

*Response*: We have now replaced "Real $N^2$" with "Real N2" in Figs. 6f, 7f, 8h and 9.

---

## Author Response (AR2)

**Response letter to the Editor**

Although you have adequately addressed the concerns/comments from Referee 2 in your response, some points necessitate additional elaboration or clarification within the manuscript. Please consider fully incorporating your replies to the comments "Lines 89-91 and 391-400 [...] " and "Lines 104-105 [...] " in the Appendices and explaining them accordingly in the main text (e.g., update the text in Discussion from Lines 394-403).

*Response*: We would like to thank the editor for this careful valuable comment. In the revision, we have carefully considered it, and incorporated the discussion of quasi-steady state and horizontally inhomogeneous stratification into the manuscript. Please find Appendix B and Appendix C in the paper and relevant discussions in the main text.

Please ensure that the colour schemes used in your maps and charts allow readers with colour vision deficiencies to correctly interpret your findings. Please check your figures using the Coblis – Color Blindness Simulator (https://www.color-blindness.com/coblis-color-blindness-simulator/) and revise the colour schemes accordingly.

*Response*: We appreciate the editor pointing out that the colour schemes in the charts might be unfriendly to the readers with colour vision deficiencies. We now check all figures using the Coblis – Color Blindness Simulator and find that Figure 9 should be definitely modified. We have now revised the colour schemes.